# Mortality and its predictors among patients treated for acute exacerbations of chronic obstructive respiratory diseases in Jimma Medical Center; Jimma, Ethiopia: Prospective observational study

**Teshale Ayele Mega**[1]*, **Zenebe Keno Anbese**[2], **Samuel D. Yoo**[3]

**1** School of Pharmacy, College of Health Sciences, Addis Ababa University, Addis Ababa, Ethiopia, **2** School of Pharmacy, College of Health and Medical Sciences, Haramaya University, Harari regional state, Haramaya, Ethiopia, **3** School of Medicine, Institute of Health Science, Jimma University, Oromia regional state, Jimma, Ethiopia

* tesh.ayu2016@gmail.com

## Abstract

### Background

Chronic obstructive pulmonary disease (COPD) and asthma exacerbations are associated with ill health, increased mortality, and health care costs. However, there is limited evidence regarding mortality and its predictors among patients treated for COPD and asthma exacerbations in low-income nations, particularly in Ethiopia.

### Methods

A-6 month prospective observational study was conducted from April 20-September 20, 2019. Data were collected on socio-demographic, baseline clinical characteristics and outcomes of asthma and COPD exacerbations. Data were entered into Epi-Data version 4.02.01 for cleaning and exported to STATA 14.0 for analysis. Kaplan-Meier (*Log-rank test*) was used to compare the baseline survival experience of the study participants and Cox proportional hazard regression analysis was conducted to determine the predictors of mortality. Adjusted hazard ratios (AHRs) with two-sided p-value <0.05 were considered statistically significant.

### Results

A total of 130 patients (60% males) were included. The median (interquartile range (IQR)) age of the study participants was 59(50–70) years. The median (IQR) survival time to death was 17.5 (10–26) days. The total proportion of in-hospital mortality was 10.78% (14/130), and the incidence rate of mortality was 2.56 per 1000 person-years. The duration of oxygen therapy $\geq$16hours/day (AHR = 6.330, 95% CI [1.092–36.679], and old age (AHR = 1.066, 95% CI [1.0001–1.136] were the independent predictors of in-hospital mortality.

**Data Availability Statement:** All relevant data are within the manuscript and its Supporting Information files.

**Funding:** The author(s) received no specific funding for this work.

**Competing interests:** The authors have declared that no competing interests exist.

**Abbreviations:** AEA, Acute exacerbation of asthma; AECOPD, Acute exacerbations of chronic obstructive pulmonary disease; AOR, Adjusted odds ratio; AHR, Adjusted hazard ratio; COPD, Chronic obstructive pulmonary disease; FEV1, Forced expiratory volume in one second; FVC, Forced vital capacity; FEF, Forced expiratory flow; GBD, Global burden of disease; GINA, Global initiatives for asthma; PEF, Peak expiratory flow; JMC, Jimma medical center.

## Conclusion

In this study, the in-hospital mortality rate was very high. Moreover, prolonged oxygen therapy (≥16hours/day) and old age were independently associated with in-hospital mortality. Therefore, special attention should be given to recipients of prolonged oxygen therapy and the elderly during hospital stay.

## Background

Asthma and Chronic obstructive pulmonary disease (COPD) are commonly called chronic obstructive respiratory diseases. They are the common conditions with a heterogeneous distribution worldwide [1]. Exacerbations of asthma and COPD are defined as an acute onset and worsening of respiratory symptoms beyond the baseline level that require a change in medication in mild cases and need emergency department (ED) visits or hospitalization in severe cases [2]. Both Asthma and COPD are the two most prevalent chronic obstructive respiratory diseases worldwide. According to the 2015 Global Burden of Disease (GBD) report, an estimated 358.2 and 174.5 million people were affected by asthma and COPD, respectively [1, 3, 4]. COPD alone is accounted for 73.3% disability-adjusted life years (DALYs), while 16.5% of the DALYs belonged to asthma [1]. COPD is responsible for nearly 800,000 hospitalizations and an estimated healthcare expenditure of $50 billion, whereas the healthcare expenditure due to asthma was $56 billion [3, 4].

Although COPD and asthma caused only 3.2 and 0.4 million deaths, respectively, by 2015 globally [1], the current regional mortality reports are quite alarming. A study conducted in Spain found 6.4% of in-hospital mortality among 972 patients admitted with acute exacerbation of COPD (AECOPD) [5]. A prospective cohort study found an in-hospital mortality rate of 8% among patients admitted with AECOPD in Netherland. This study also reported long-term use of oral corticosteroids, higher PaCO2, and older age as risk factors for in-hospital mortality [6]. A systematic review unveiled 3.6% of short-term mortality, 31% of long-term mortality (up to 2year), and 29% intensive care unit (ICU) mortality among patients admitted with AECOPD [7]. Furthermore, a multicentre cohort study reported a cumulative 180-day mortality of 37.9% among AECOPD and asthma patients admitted to ICU [8]. Another study found a 23.1 per 1000 person-years and 7.79 per 1000 person-years incidence rate of mortality corresponding to AECOPD and asthma, respectively [9].

In Ethiopia, national data regarding burden and different aspects of acute exacerbations of chronic obstructive respiratory diseases are limited. However, there are some disaggregated studies conducted in different parts of the country. In one study, the prevalence of asthma was 3.5% -9.1% and that of COPD was 4% [10]. About 0.6% of deaths, both in urban and rural Ethiopia, were caused by asthma. COPD related mortality in the southern and central part of Ethiopia was 5.2% and 3%, respectively [11].

Moreover, a recent study indicated the previous and family history of chronic obstructive respiratory disease, previous dusty working environment, and ever smoking were the main risk factors for chronic obstructive respiratory symptoms and its exacerbation [12]. Several risk factors like cigarette smoking, ambient particulate matter, household air pollution, occupational particulates, ozone, and second-hand smoke can cause COPD. Indoor smoke from solid fuels, outdoor air pollution, and occupational asthmagens were risks quantified for asthma in GBD [1]. Exacerbations due to lack of early prevention of these risk factors and comorbidity were the main cause of hospital admission [1, 3, 4].

Therefore, this study was aimed to assess mortality and its predators among patients admitted to Jimma medical center (JMC) and treated for acute exacerbations of chronic obstructive respiratory diseases, particularly asthma and COPD.

## Methods

### Study area and period

This study was conducted at the medical ward, Emergency department (ED), and intensive care units (ICUs) of Jimma Medical Center (JMC). JMC is located in Jimma town, Jimma Zone, Oromia Region, Southwest Ethiopia. It is about 346 km away from the Ethiopian capital, Addis Ababa. JMC is one of the oldest public hospitals found in the south-western part of the country. It is the only medical center providing the most advanced level of referral care in this part of the country and has a catchment population of 15 million people. The hospital has more than 800 beds, 1600 staff members, and 32 intensive care units [13].

Jimma Medical Center has several specialty clinics. One of the speciality clinics is the chest clinic. This clinic has an outpatient department (OPD) and a pulmonology unit, the major source of patients for this study. This study was conducted from April 20-September 20, 2019.

### Study design, population and sample size determination

A prospective observational study was conducted among patients admitted with chronic obstructive respiratory disease. We included all adult (age >18years) patients with spirometry confirmed diagnosis of asthma and COPD, patients with an obstructive pattern, and patients admitted with acute exacerbations of asthma and COPD. Patients receiving mechanical ventilation, misdiagnosed and cardiac asthmatics were excluded. The study was approved by the Institutional Review Board (IRB) of Jimma University and labelled with an IRB number of IHRPGD/549/109. A signed informed consent was taken from the patients if they were able to communicate and otherwise consent was requested from their caregivers. During data collection, confidentiality was ensured and thus, the name of the patient was not recorded in the data collection check list. Since a limited number of patients were encountered during the study period, we included all patients who fulfilled the eligibility criteria and no special sampling technique employed. Patients were followed prospectively for 6-months and the data of 130 (asthma = 59) and COPD = 71) patients were analyzed.

### Study variables and data collection procedure

In-hospital mortality was considered as an outcome variable. Data regarding the following variables were collected from active patient follow-up charts and patient interviews.

1. **Demographic data**

   a. Anthropometric data including age, sex, weight, height, and body mass index (BMI) were collected from the patients' chart. Admission history, dates of index admission and discharge were also collected from the patients' chart.

   b. Data regarding social histories such as smoking status, occupation, marital status, and availability of home care services were acquired from patient interviews.

   c. History of treatment related data like duration of oxygen therapy, antibiotics regimen and the dose of inhaled or systemic corticosteroids were obtained from the patient interview and medication record charts.

2. **Clinical variables**

a.  Baseline clinical characteristics data such as vital signs, comorbidities, clinical presentations, and lung function tests (PEF, FEV1, and FEV1/FVC1) were taken from patients' active follow-up charts. The lung function test was conducted after the patient is being stabilized at the emergency department. For all patients, the lung function test was performed by appropriately-trained technician using CareFusion-microlab3500, V1.34: CAR36ML3500SR4 spirometry).

b.  Laboratory data such as hemoglobin, haematocrit, serum albumin, renal function tests, and total white blood cell count were extracted from active patients' follow-up charts. Complete blood counts were measured using hematologic analyzers: XT-1800i (Sysmex, Japan), KX-21 N™ (Sysmex, Japan), and Cell-Dyn 18001(Abbot, USA). Serum albumin and renal function tests (serum creatinine, and blood urea nitrogen) were measured using chemistry analyzers: ABX Pentra 400(Horiba, USA), Dirui DR-7000D (DIRUI, Changchun, China) and HumaLyzer 3000 (HUMAN, Wiesbaden, Germany).

3. **Disease severity and outcome measurment**

The severity of COPD and asthma was evaluated using the global obstructive lung disease (GOLD) and global initiative for asthma (GINA) guidelines, respectively [14, 15]. Adherence to asthma and COPD medications inhalation techniques were assessed by passive observation. In this regard, patients were expected to demonstrate the 10 steps of the inhalation technique, the validated questionnaire. In-hospital mortality data were acquired from the patients' discharge summary, as described and confirmed by the caring physician.

4. **Tool validation and data quality assurance**

The data collection tool was submitted to a team of pulmonlogists working in Jimma medical center. They thoroughly discussed on its details and provided some comments. After including the comments, the tool was sent to the psychometrician for further evaluation. An amendement was also made based on the psychomerician's recommendation, and a pre-test was conducted on 10 randomly selected eligible patients. The pre-test data was double entered into and cleaned with spread sheet. Later on, a principal component analysis was conducted by statistician to identify the factor loadings; factors that the tool can actually measure. Finally, the internal consistency was checked and revision was made to the final data collection tool. After the completion of the data collection tool, four data collectors (two Pharmacists with a Bachelor degree in pharmacy and two clinical nurses with a Bachelor of Science degree) and four supervisors (general practitioners) were hired. They were given two days of intensive training on the data collection tool, and general procedures. The role of the supervisors was supervising the data collectors and facilitating the daily activities.

5. **Diagnostic procedures for acute exacerbations of chronic respiratory diseases**

All spirometry measurements were made using identical care fusion spirometry (V1.34: CAR36ML3500SR4 (brand: CareFusion-microlab3500 spirometry). Patients were asked to omit short-acting inhaled bronchodilators for 4–6 hours, and long-acting oral and inhaled agents for 12 hours before receiving lung function testing. Lung function test was taken at baseline and repeated 15minutes after the patients were challenged with a Salbutamol puff of 400mcg (short-acting beta- agonist (SABA)) [14].

The diagnosis of COPD was confirmed if the post-bronchodilator (400mcg of SABA) peak flow measured after 10–15 minutes was increased by less than 12% or 200ml from the baseline of PEF or FEV1(pre-bronchodilator) or evaluated FEV1 was less than or equal to 80% predicted for age, height, and sex, and FEV1/FVC ratio was less than 0.7. Acute exacerbation of COPD was diagnosed if the patients are presented with progressive dyspnoea, chronic cough and increased in sputum prulence/production [14].

The diagnosis of asthma was confirmed if the post-bronchodilator (400mcg of SABA) peak flow measured after 10–15 minutes was increased by 12% or 200ml from the baseline of PEF or FEV1 (pre-bronchodilator) and FEV1/FVC ratio was less than 0.7 and oxygen saturation $\geq$90. The presence of cough, wheezing, increased shortness of breath and presence of risk factors, were the onfirmatory diagnosis for acute exacerbation of asthma [15].

## Data processing and analysis

Data were entered into Epi-Data 4.02.01 for cleaning and exported to STATA 14 for analysis. Descriptive analysis was performed and results were presented in text, tables, and charts. Continuous variables were reported using median and interquartile ranges. The baseline survival experience of the patients was checked by Kaplan-Meier (*Log-rank test*).

A chi-square test was performed to check the difference in baseline characteristics and adequacy of cells for Cox regression. The Cox regression model assumption of proportional hazards was checked by testing the interaction of covariates with time. Bivariate Cox regression was performed to pick variables for multivariable Cox regression. Variables with p-value < 0.25 in bivariate regression were considered for multivariable Cox regression. Multivariate Cox regression was performed to identify independent predictors of in-hospital mortality. The hazard ratio (HR) was used as a measure of the strength of association and p-value < 0.05 was considered to declare statistical significance.

## Results

During the study period, 162 patients were admitted with the diagnosis of acute exacerbations of chronic obstructive respiratory disease and followed prospectively for 6-months. Of the 162 eligible patients, 32 were excluded due to several reasons and 130 (asthma = 59, COPD = 71) patients were included in the final analysis (Fig 1).

## Description of socio-demographic and clinical characteristics

Of the 130 eligible patients, 78(60%) were males. Males contributed to the most (78.6%) of the deaths. The median age of the study partcipants was 59(IQR, 50–70) years and 68(52.3%) of the patients were from rural areas. More than half, 73(56.15%) of the patients had no formal education. About 64.3% of the deaths were from this group. Most (60.76%) of the patients were farmers and housewives. Seventy (59.32% asthmatics) patients had a history of living with pet animals (cats, dogs), and most (71.4%) of the deaths were reported from this group.

Most, 113(86.92%) of the study participants were married. Twenty-eight (21.54%) patients were active smokers and most (85.7%) of the deaths were reported among patients with a history of smoking. About 25(35.21%) of the smokers were diagnosed with COPD, and only 3(5.08%) of the smokers were asthmatic patients. Twenty-three of the smokers used 1–10 cigarettes/day. At baseline, 39(30%) of the participants had a low body mass index. i.e. (BMI<18.5kg/m$^2$), one of the contributors for the baseline difference among the patients (p$\leq$0.001) (Table 1).

The majority, 79(60.77%) of patients had a previous history of hospital admission. Nearly two-thirds (64.3%) of the deaths were reported from this group. Seventeen (5.6%) patients had history of repeated ($\geq$ 3 attacks/year) hospital admissions. According to GINA guideline, 34 (29.06%) patients encountered severe night attacks, i.e. 7attacks/week (asthma = 18, COPD = 16). However, fewer (7.1%) deaths were reported from this group. Acute exacerbation of COPD was responsible for the most (71.4%) of the deaths. Sixty-one (46.92%) patients (asthma = 20, COPD = 41) had chronic comorbidities and 78.6% of the deaths were reported from this group. Thirty-four (47.89%) patients had pulmonary hypertension and 23(32.39%)

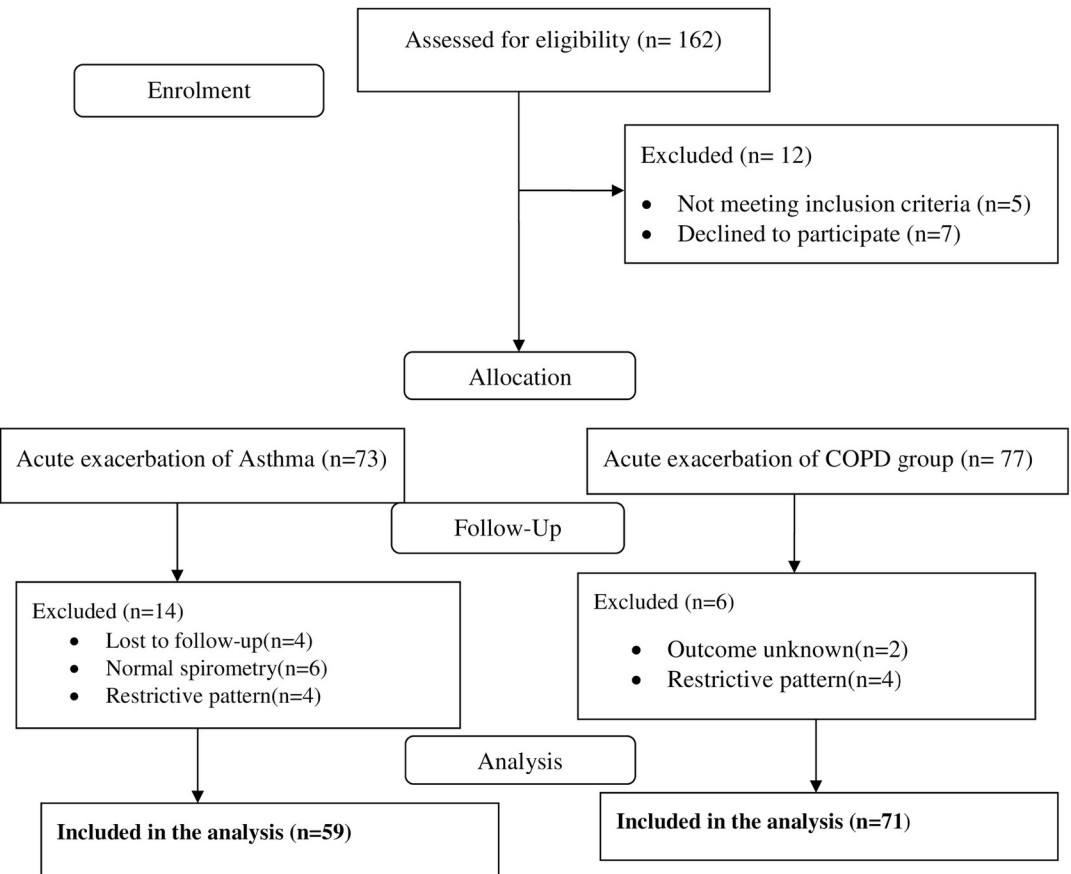

**Fig 1. Sample recruitment diagram of study participants at JMC from April 20-September 20, 2019.**

patients had heart failure or (IHD). Pulmonary hypertension (9-deaths, p = 0.003) and Heart failure (IHD) (7-deaths, p = 0.008) were tend to impact patients' status (Table 2).

## Description of medications and laboratory findings

Short acting beta-agonsts (SABA) and oral or inhalaional corticosteroids were the major components of the care. In this study, Salbutamol puff and prednisolone were prescribed to 128 (asthma = 58, COPD = 70) and 81 (asthma = 35, COPD = 46) patients, respectively. These medications were also the backbone of the post-hospital care for most of the patients. Fifty-one (39.23%) patients had previous exposure to oxygen therapy. The duration of oxygen therapy $\geq$16hours/day was found to impact the patient's status (p$\leq$0.001). Patients remaining on oxygen for $\geq$16hours/day were discharged with oxygeen after being linked to the out patient department for regular follow-up. Antibiotics were also prescribed as an adjunctive therapy for acute exacerbations during hospital stay and patients may take one or combinations of antibitiotics based on the existing risk factors for multiple or resistance etiologies. Sometimes antimicrobial regimens may be modified based on patients' response to initial therapy. In this study, 97 (13 died) patients received ceftriaxone and 91 (13 died) patients received Azithromycin. Exposure to ceftriaxone (p = 0.097) and Azithromycin (p = 0.048) was contributed to the baseline difference. But, this association was not replicated with logestitic regression. The full description of the in-patients medications used for the management of acute exacerbations of asthma and COPD is annexed (S1 Annex).

**Table 1. Baseline socio-demographic characteristics of the study subjects at JMC, April 20-September 20, 2019.**

| Socio-demographic variables (n = 130) | | Patient status | | $\chi^2$ p-value |
|---|---|---|---|---|
| | | Died | Live | |
| Gender | Male | 11 | 67 | 0.133 |
| | Female | 3 | 49 | |
| Age Median (IQR) | 59 (50–70 years) | 14 | 116 | 0.614 |
| Body mass index (BMI) | <18.5kg/m2 | 14 | 35 | p≤0.001 |
| | ≥18.5kg/m2 | 0 | 81 | |
| Marital status | Single | 0 | 7 | 0.633 |
| | Married | 13 | 100 | |
| | Widowed | 1 | 9 | |
| Residence | Urban | 4 | 58 | 0.129 |
| | Rural | 10 | 58 | |
| Educational level | Can't read & write | 9 | 48 | 0.176 |
| | Primary | 2 | 44 | |
| | Post-primary | 3 | 24 | |
| Occupation | Employed | 3 | 48 | 0.157 |
| | Farmer | 8 | 37 | |
| | Housewife | 3 | 34 | |
| Home status | With family | 14 | 109 | 0.345 |
| | Alone | 0 | 7 | |
| Live with pet animals | Yes | 10 | 60 | 0.162 |
| | No | 4 | 56 | |
| Smoking status | Active smokers | 6 | 22 | 0.121 |
| | Non-smokers | 6 | 72 | |
| | Ex-smokers | 2 | 22 | |
| Active smokers (estimated cigarette/day) | 1–10 cigarette/day | 5 | 18 | 0.859 |
| | 11–20 cigarette/day | 1 | 3 | |
| | 21–30 cigarette/day | 0 | 1 | |

IQR: Inter quartile range.

Although elevated baseline hemoglobin was reported in 20 (15.38%) (asthma = 2, COPD = 18) patients, 57.14% of the deaths were reported from patients with normal baseline hemoglobin levels. Fifty nine (45.38%) patients presented with high white blood cells (asthma = 28, COPD = 31). Serum creatinine was increased in about 21(asthma = 6, COPD = 15) patients and contributed to 57.1% of the deaths. Elevated blood urea nitrogen was reported in 16 (asthma = 5, COPD = 11) patients (Table 3).

At admission, all eligible patients were subjected to lung function testing. The ratio of forced expiratory volume in one second to forced vital capacity (FEV1/FVC) was <0.7 in 130 patients showing an obstructive pattern. The pre and post-bronchodilator test results were depicted in Table 4. Although, the pre and post-bronchodilator test results of FEV1/FVC had a mean deference of 4.4% and 2.8% (p = 0.829), respectively, there was no statistically significant difference in patients' status.

## Measurements of inhalation technique adherence

For the 130 patients analyzed, 10 step-wise inhalation techniques were provided to check for patients' adherence to the effective inhalation techniques. Patients were found to be adherent if they performed all steps without making any mistake. Based on these criteria, only 14(10.77%)

**Table 2. Baseline clinical characteristics of subjects at JMC, April 20-September 20, 2019.**

| Clinical characteristics (n = 130) | | Patient status | | $\chi^2$ p-value |
|---|---|---|---|---|
| | | Died | Live | |
| History of hospital admission | Yes | 9 | 70 | 0.775 |
| | No | 5 | 46 | |
| Number of hospital admission per year | Never | 5 | 40 | 0.843 |
| | Once | 2 | 22 | |
| | Twice | 5 | 33 | |
| | ≥ 3 attacks/year | 2 | 15 | |
| Frequency of night attacks | 3–4 attacks/month | 13 | 70 | 0.054 |
| | 7 attacks/week | 1 | 33 | |
| Admission diagnosis | Asthma | 4 | 55 | 0.181 |
| | COPD | 10 | 61 | |
| Limit daily activity | Yes | 12 | 48 | 0.002 |
| | No | 2 | 68 | |
| Chronic comorbidities | Yes | 9 | 14 | 0.168 |
| | No | 5 | 116 | |
| Heart failure or IHD | Yes | 7 | 22 | 0.008 |
| | No | 7 | 94 | |
| Hypertension | Yes | 1 | 16 | 0.486 |
| | No | 13 | 100 | |
| Diabetes mellitus | Yes | 0 | 4 | 0.480 |
| | No | 14 | 112 | |
| Cardiac arrhythmia | Yes | 1 | 6 | 0.758 |
| | No | 13 | 110 | |
| Depression | Yes | 0 | 1 | 0.727 |
| | No | 14 | 115 | |
| Pulmonary Hypertension | Yes | 9 | 30 | 0.003 |
| | No | 5 | 86 | |
| Renal failure | Yes | 3 | 5 | 0.012 |
| | No | 11 | 111 | |
| Cancer | Yes | 0 | 4 | 0.480 |
| | No | 14 | 112 | |
| Deep venous thrombosis | Yes | 1 | 1 | 0.071 |
| | No | 13 | 115 | |

IHD: Ischemic heart disease, COPD: Chronic obstructive pulmonary disease.

patients were adherent to the inhalation techniques. The remaining, 116 (89.23%) patients missed at least one step. As steps go up, patients' ability to remember and perform effectively was decreased and most, 91(70%) of the patients cannot perform the last step correctly (Fig 2).

## Mortality

In our study, the rate of in-hospital mortality among patients admitted with acute exacerbation of chronic obstructive respiratory disease was 10.78/100 patients. Four 4(3.08%) deaths were due to acute exacerbation of asthma (AEA) and 10(7.69%) deaths were linked to AECOPD. The incidence rate of mortality for acute exacerbation of chronic obstructive respiratory disease was 2.56 per 1000 person-years; 95% CI [1.518–4.328] with the median (IQR) survival time of 17.5 (10–26) days. The incidence rate of mortality corresponding to AECOPD was

**Table 3. Baseline drug-related and laboratory findings of the study participants at JMC, April 20 -September 20, 2019.**

| In-patient medications and blood parameters (n = 130) | | Patient status | | $\chi^2$ p-value |
|---|---|---|---|---|
| | | Died (n = 14) | Live (n = 116) | |
| Ceftriaxone 1gm IV twice daily | Yes | 13 | 84 | 0.097 |
| | No | 1 | 32 | |
| Azithromycin 500mg pd daily | Yes | 13 | 78 | 0.048 |
| | No | 1 | 38 | |
| Vancomycin 1gm bid | Yes | 1 | 4 | 0.497 |
| | No | 13 | 112 | |
| History of oxygen therapy | Yes | 7 | 44 | 0.382 |
| | No | 7 | 72 | |
| Duration on oxygeen per 24hours | <16hrs | 2 | 76 | p≤0.001 |
| | ≥16hrs | 12 | 4 | |
| High dose salbutamol puff | Yes | 10 | 90 | 0.605 |
| | No | 4 | 20 | |
| High dose salmeterol/formaterol puff | Yes | 0 | 1 | 0.727 |
| | No | 14 | 115 | |
| Beclomethasone Puff | Yes | 4 | 14 | 0.091 |
| | No | 10 | 102 | |
| Prednisolone tablet | Yes | 11 | 70 | 0.184 |
| | No | 3 | 46 | |
| Hydrocortisone injection | Yes | 7 | 31 | 0.070 |
| | No | 7 | 85 | |
| Budesonide + formoterol inhalations | Yes | 1 | 4 | 0.497 |
| | No | 13 | 112 | |
| Inhalational technique adherence | Yes | 3 | 11 | 0.173 |
| | No | 11 | 105 | |
| Haemoglobin in g/dl | High | 4 | 16 | 0.093 |
| | Normal | 8 | 77 | |
| | Low | 2 | 5 | |
| White blood cell count | High | 7 | 52 | 0.396 |
| | Normal | 7 | 43 | |
| | Low | 0 | 3 | |
| Serum creatinine | High | 8 | 13 | p≤0.001 |
| | Normal | 4 | 35 | |
| | Low | 1 | 16 | |
| Blood urea nitrogen | High | 7 | 9 | p≤0.001 |
| | Normal | 4 | 49 | |
| | Not measured | 3 | 56 | |
| | Low | 0 | 2 | |

3.434 per 1000 person-years; 95% CI [1.848–6.382) with the median (IQR) survival time of 21.5 (11–26) days, while that of AEA was 10.058 per1000 person-years; 95% CI [3.775–26.798] with the median (IQR) survival time of 10 (6.5–25.5) days. The overall survival among patients with AEA and AECOPD was not statistically different (Log-rank p = 0.73) (Fig 3).

## Predictors of mortality

Bivariate Cox-regression analysis indicated low body mass index (BMI<18.5kg/m2), limited daily activity, increasing frequency of night attacks, non-adherence, old age, prolonged use of

**Table 4. Summary of lung function tests of the study participants at JMC, April 20-September 20, 2019.**

| Lung function test result (n = 130) | | | | | | | | | | |
| --- | --- | --- | --- | --- | --- | --- | --- | --- | --- | --- |
| Pre bronchodilator test | | | | | | Post bronchodilator test | | | | |
| | Died (n = 14) | | Live (n = 116) | | | Died (n = 14) | | Live (n = 116) | | |
| Parameters | Mean | SD. | Mean | SD. | $\chi^2$- p | Mean | SD. | Mean | SD. | $\chi^2$—p |
| FVC(L) | 1.138 | 0.459 | 1.454 | 0.633 | 0.932 | 1.197 | 0.634 | 1.545 | 0.685 | 0.866 |
| FVC predicted (%) | 43.143 | 16.617 | 53.724 | 21.705 | 0.951 | 46.0 | 20.550 | 56.405 | 21.924 | 0.901 |
| FEV1 (L) | 0.563 | 0.167 | 0.826 | 0.432 | 0.999 | 0.586 | 0.183 | 0.899 | 0.433 | 0.893 |
| FEV1 predicted (%) | 31.286 | 13.809 | 42.638 | 18.352 | 0.98 | 31.286 | 13.809 | 42.638 | 18.352 | 0.728 |
| FEV1/FVC(ratio) | 0.528 | 0.138 | 0.572 | 0.1097 | 0.944 | 0.561 | 0.166 | 0.589 | 0.114 | 0.956 |
| PEF | - | | - | | | 125.615 | 134.568 | 127.876 | 78.366 | 0.950 |

FVC: Forced vital capacity, FEV1: Forced expiratory volume in one second, PEF: Peak expiratory flow, SD: standard deviation.

oxygen therapy per 24hrs, prolonged use of high dose Inhaled corticosteroids (ICS) and frequent use of hydrocortisone injection were associated with in-hospital mortality (p<0.25) (Table 5).

However, after adjusting for all confounders with multivariate Cox regression model, prolonged use of oxygen per 24hours and old age became independent predictors of mortality. Accordingly, being on oxygen therapy for ≥16hours per day increased the hazard of death by 6.8(AHR = 6.833, 95%CI [1.300–35.905]. Similarly, a unit increment in age increased the hazard of death by 6.6% (AHR = 1.066, 95%CI, [1.00005–1.13582] (Table 5).

## Discussion

This study summarized mortality and its predictors among admitted patients who received treatment for acute exacerbations of chronic obstructive respiratory disease, particularly patients with acute exacerbations of asthma (AEA) and chronic obstructive pulmonary disease (AECOPD). The study found the overall mortality of 10.78% due to chronic obstructive respiratory disease. The corresponding incidence rate of mortality was 2.56 per 1000 person-years; 95% CI [1.518–4.328]. The incidence rate of mortality was 3.434 per 1000 person-years, 95% CI [1.848–6.382), and 10.058 per1000 person-years, 95% CI [3.775–26.798] for AECOPD and

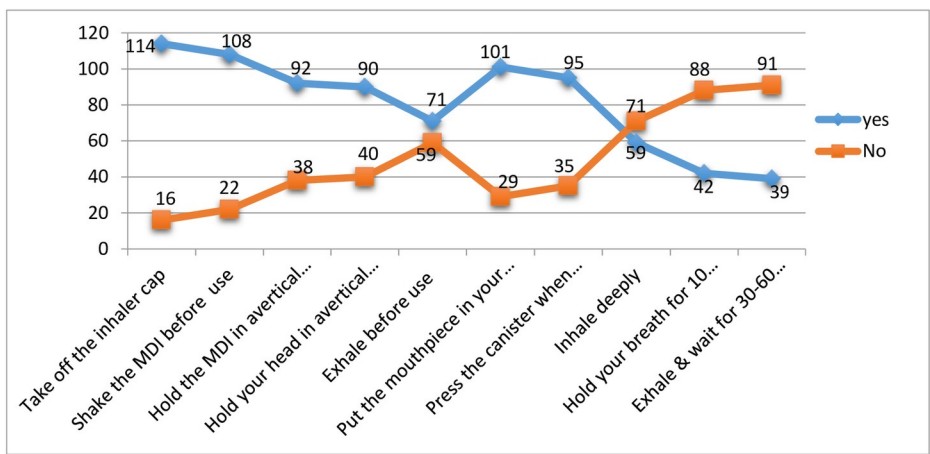

**Fig 2. Adherence to the instructions of metered-dose inhalation (MDI) techniques among study participants at JMC, April 20-September 20, 2019.**

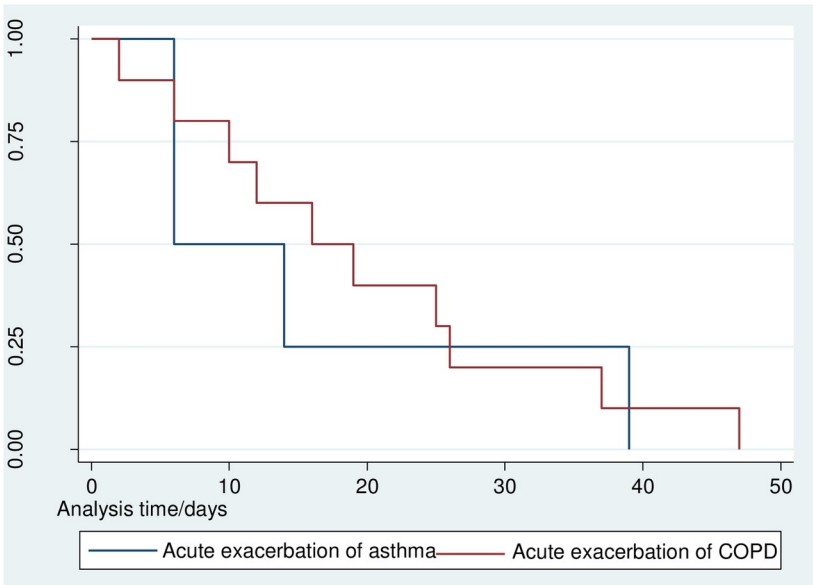

**Fig 3. Survival estimates for in-hospital mortality among patients admitted with acute exacerbations of asthma and COPD at JMC, April 20-September 20, 2019.**

AEA, respectively. The overall median (IQR) survival time for obstructive respiratory disease was 17.5 (10–26) days. The median (IQR) survival times for AECOPD and AEA were 21.5 (11–26) and 10 (6.5–25.5) days, respectively (p = 0.73). Prolonged use of oxygen therapy (≥16hrs) and old age were independent predictors of in-hospital mortality.

In-hospital mortality of 26.3% was reported by Sana et al. among 259 patients admitted with similar conditions [16]. This finding was higher compared to the present study (26.3% vs. 10.78%). The deviation could be due to the difference in sample size (259 vs. 130) and duration of the studies (1year vs. 6 months). Moreover, Wildman et al. [8] found a cumulative 180-days

**Table 5. Crude and adjusted Cox proportional hazard regression for predictors of mortality of JMC, April 20- September 20, 2019.**

| Variables | | Status | | CHR [95% CI] | p-value | AHR[95% CI] | p-value |
|---|---|---|---|---|---|---|---|
| | Category | Died | Live | | | | |
| Age (years) | Median | 70 | 57.5 | 1.03 [0.980–1.086] | 0.236 | 1.066[1.00005–1.13583] | 0.050 |
| Body mass index (kg/m2) | = >18.5 | 1 | 80 | 1 | | 1 | |
| | <18.5 | 13 | 36 | 8.896[1.150–68.809] | 0.036 | 2.213[0.199–24.633] | 0.518 |
| Adherence | No | 11 | 105 | 0.40 [0.107–1.536] | 0.184 | 0.837 [0.172–4.068] | 0.825 |
| | Yes | 3 | 11 | 1 | | 1 | |
| Prolonged use of oxygen therapy | <16hrs | 2 | 75 | 1 | | 1 | |
| | ≥16hrs | 12 | 5 | 10.597 [2.342–47.948] | 0.002 | 6.833[1.300–35.905] | 0.023 |
| Bechlomethasone inhalation | No | 1 | 72 | 1 | | 1 | |
| | yes | 13 | 44 | 6.269 [0.789–49.27] | 0.081 | 0.893[0.074–10.795] | 0.93 |
| Hydrocortisone injections | No | 6 | 81 | 1 | | 1 | |
| | yes | 8 | 35 | 2.857 [0.988–8.258] | 0.053 | 2.956[0.729–11.984] | 0.129 |
| Frequency of night attacks | 3–4 attacks /month | 13 | 70 | 1 | | 1 | |
| | 7attacks /week | 1 | 33 | 0.235[0.031–1.808] | 0.164 | 0.241[0.023–2.494] | 0.233 |
| Limit daily activity | No | 2 | 68 | 1 | | 1 | |
| | Yes | 12 | 48 | 3.175[0.703–14.336] | 0.133 | 3.391[0.571–20.124] | 0.179 |

in-hospital mortality of 37.9% for obstructive airway disease. This disagreement with the current study might be due to the sample size (832 vs. 130) and inclusion of patients from Intensive care unit (ICU) in the former study. Our finding also differs from the study by Baarnes et al. [9], which reported 10.9per 1000 persons-years; 95% CI [3.64–33.33] of overall mortality. In a similar study, the incidence rates of mortality for AECOPD and AEA were 23.1per1000 person-years; 95% CI [21.82–24.42] and 7.79per 1000 person-years; 95%CI [6.64–9.13], respectively. The difference might be due to the sample size, duration of study, and differences in population genetics.

Study by Meservey and his coleagues [17] found in-hospital mortality of 15(7%) among 202 (COPD = 46%, Asthma = 10%) admitted patients. This finding was nearly in agreement with the current study. Dwyer-Lindgren et al. [18] reported a lower (52.9 per100, 000) incidence rate of in-hospital mortality among patients with chronic obstructive respiratory disease. Improved patient care, well-equipped care facility with better access to inhaled corticosteroids might justify this difference. A similar finding of AE asthma mortality (10.4 per 1000 person-years) was reported by S. Vandentorre et al. [19] and Italian (11.0per 100,000 person-years; 95% CI[8.3–14.7]) researchers [20]. However, Watson et al. [21] provided a concerning figure (2,878 per100,000 person-years; 95% CI [2091–3857]) of in-hospital mortality associated with AEA.

Another study [22] reported a similar finding with that of ours regarding in-hospital mortality of AECOPD, i.e. 4.3 per1000 person-years (p<0.001). However, a study from Canada showed higher (22.5 per 1000 person-years) incidence rate of in-hospital mortality [23]. The difference might be due to the geographical location and duration of the study period (10 years versus 6 months).

Oxygen dependence was one of the major determinants of in-hospital mortality in patients with acute exacerbations of chronic obstructive respiratory diseases. Accordingly, patients receiving oxygen therapy for ≥16hourrs per 24hrs had 6.8 times higher hazard of mortality (AHR = 6.833, 95% CI [1.3003–35.905]). Similarly, as age increases by one unit, the hazard of mortality was fund to be increased by 6.6% (AHR = 1.066, 95% CI [1.001–1.136]).

A consistent finding was reported from Malaysia showing older age (AHR = 2.53; 95% CI [1.29–4.92]) and long term oxygen therapy (AHR = 2.78; 95% CI [1.54–5.02]) were independently associated with in-hospital mortality [16]. Meservey et al. [17] also discovered a similar fact as older age (AOR = 1.32; 95%CI [1.13–1.54]) and long-term oxygen therapy (AOR = 4.03; 95% CI [1.89–8.57]) were independent predictors of in-hospital mortality. An old age (AOR = 3.24; 95% CI [1.95–5.39]) was also implicated to increase in-hospital mortality by findings from Tokyo-Japan [24] and Spain (AHR = 1.62; 95% CI [1.80–3.25]) [25]. Moreover, Wildman et al. [8] (AOR = 1.39; 95% CI [1.92–2.09]) and Fuso et al [26] (AOR = 1.07; 95% CI [1.04 to 1.11]) revealed the impact of old age on in-hospital mortality.

Finally, our study is not without limitations. The shorter duration of the study, and the inclusion of small number of patients may affect statistical power of the study. Additionally, being a single-center study and exclusion of patients requiring mechanical ventilation may limit the study's ability to detect the actual risk factors of in-hospital mortality among the study particiants. More importantly, merging the outcomes of two chronic obstructive conditions, namely; asthma and COPD, may deter the generalizability of the study findings.

## Conclusion

In this study, in-hospital mortality was very alarming. Long term oxygen therapy (≥16hourrs/day) and old age were significantly associated mortality. Providing special attention during treatment and strict follow-up after discharge is mandatory. Since the post discharge follow-up will be continued at the out-patient department of the pulmonary unit, it might be very

difficult to reach them during acute exacerbations due to the scarcity of ambulance services. However, whether the above issues could impact mortality among the study population should be further investigated. Thus, we call for a long term study with adequate sample size to find out the loopholes of this study and ensure its generalizability. Of note, the finding of this study must be interpreted cautiously as it did not investigate the predictors of acute exacerbations of asthma and COPD separately.

## Supporting information

**S1 Annex. Description of in-patient medications for the management of Asthma and COPD exacerbations.**
(DOCX)

**S1 File. Data collection tool.**
(DOCX)

**S2 File. Mortality of AECRD (STATA).**
(DTA)

**S3 File. STATA out put (final model).**
(PDF)

## Acknowledgments

We would like to thank the study participants, pharmacists, nurses, and physicians at JMC for their indispensable cooperation during the acquisition of data. Lastly, we also like to extend our heartfelt gratitude to Mr. Getandale Zeleke Negera for his contribution to revising the final version of the manuscript.

## Author Contributions

**Conceptualization:** Teshale Ayele Mega, Zenebe Keno Anbese.

**Data curation:** Zenebe Keno Anbese.

**Formal analysis:** Teshale Ayele Mega, Zenebe Keno Anbese.

**Investigation:** Samuel D. Yoo.

**Methodology:** Teshale Ayele Mega, Zenebe Keno Anbese, Samuel D. Yoo.

**Resources:** Zenebe Keno Anbese.

**Software:** Teshale Ayele Mega, Zenebe Keno Anbese.

**Supervision:** Teshale Ayele Mega, Samuel D. Yoo.

**Validation:** Zenebe Keno Anbese, Samuel D. Yoo.

**Visualization:** Zenebe Keno Anbese.

**Writing – original draft:** Teshale Ayele Mega.

**Writing – review & editing:** Teshale Ayele Mega, Zenebe Keno Anbese, Samuel D. Yoo.

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
