## [Decision Letter · Decision Letter 0]

12 May 2020

PONE-D-20-02427

Mortality and its predictors among patients admitted with acute exacerbation of chronic obstructive respiratory diseases in Jimma Medical Center: Prospective cohort study

PLOS ONE

Dear Teshale Mega Ayele,

Thank you for submitting your manuscript to PLOS ONE. After careful consideration, we feel that it has merit but does not fully meet PLOS ONE’s publication criteria as it currently stands. Therefore, we invite you to submit a revised version of the manuscript that addresses the points raised during the review process.

We would appreciate receiving your revised manuscript by Jun 26 2020 11:59PM. To enhance the reproducibility of your results, we recommend that if applicable you deposit your laboratory protocols in protocols.io, where a protocol can be assigned its own identifier (DOI) such that it can be cited independently in the future. For instructions see: http://journals.plos.org/plosone/s/submission-guidelines#loc-laboratory-protocols

We look forward to receiving your revised manuscript.

Kind regards,

Muhammad Adrish

Academic Editor

PLOS ONE

2. Thank you for including your ethics statement:  The full protocol of this study was submitted for Institutional Review Board (IRB: IHRPGD/549/109) of JUMC, for ethical approval. After informing the overall concern of the study and confidentiality, written informed consent was taken written letter sheets; for those patients with problem of speaking,

consent was requested from relatives.  

Please amend your current ethics statement to confirm that your named institutional review board or ethics committee specifically approved this study.

3. Please correct your reference to "p=0.000" to "p<0.001" or as similarly appropriate, as p values cannot equal zero.

4. Please include additional information regarding the survey or questionnaire used in the study and ensure that you have provided sufficient details that others could replicate the analyses. For instance, if you developed a questionnaire as part of this study and it is not under a copyright more restrictive than CC-BY, please include a copy, in both the original language and English, as Supporting Information. Moreover, please include more details on how the questionnaire was pre-tested, and whether it was validated.

6. Please amend either the abstract on the online submission form (via Edit Submission) or the abstract in the manuscript so that they are identical.

7. Your ethics statement must appear in the Methods section of your manuscript. If your ethics statement is written in any section besides the Methods, please move it to the Methods section and delete it from any other section. Please also ensure that your ethics statement is included in your manuscript, as the ethics section of your online submission will not be published alongside your manuscript.

Reviewers' comments:

Reviewer's Responses to Questions

**Comments to the Author**

1. Is the manuscript technically sound, and do the data support the conclusions?

Reviewer #1: Partly

2. Has the statistical analysis been performed appropriately and rigorously? 

Reviewer #1: Yes

3. Have the authors made all data underlying the findings in their manuscript fully available?

Reviewer #1: Yes

4. Is the manuscript presented in an intelligible fashion and written in standard English?

Reviewer #1: No

5. Review Comments to the Author

Reviewer #1: The manuscript, "Mortality and its predictors among patients admitted with acute exacerbations of chronic obstructive respiratory diseases in Jimma Medical Center: Prospective cohort study", by T. Mega and colleagues reports the outcomes of a cohort of patients admitted with COPD exacerbations to a referral hospital in a low-resource country. The study is descriptive, but still adds to our understanding of the outcomes for COPD patients in different countries. However, the study has several important limitations, which are outlined below:

1. There are several grammatical and spelling errors throughout the manuscript, which makes the manuscript difficult to read. For example, the third sentence of the fourth paragraph of the Background reads, "An indoor (like smoking), outdoor air pollution and occupational asthma gens were risks quantified for asthma in GBD". This sentence appears to be missing a word or phrase. Additionally, "gens" is a mistake, and the authors' intent is not clear regarding what word is meant to be in this sentence. The manuscript should be carefully reviewed and these errors corrected.

2. The study population is very small given that the hospital serves a population of 15 million. The authors should provide reasons for this.

3. The authors should provide specific criteria as to how the underlying diagnosis of asthma or COPD was made in the cohort patients.

4. The authors should provide some context for the demographic information of their cohort. Specifically, they should indicate whether there are specific differences between the patients and the rest of the communities served by Jimma Hospital, or if the patient cohort is generally representative of the surrounding areas.

5. There should be more details about the in-patient therapy received by the study cohort. Did any of the patients receive mechanical ventilation? What was the dose range and duration of steroid therapy used for treating the COPD/asthma exacerbations? This information will allow the interested reader to understand the differences in therapy received by the patients in the Jimma cohort and other COPD populations.6

6. The authors should summarize the post-hospital follow up provided to the patients. Was there a systematic post-discharge educational program given to the patients? Is there a community-based nursing follow up available?

7. The finding that patients with comorbid disease were less likely to die during follow up compared to patients without comorbidities is unexpected. The authors should enlarge their Discussion of this finding to include their assessment as to the reasons for this.

Reviewer 2

This is an interesting study and the objective of the study are well stated. In the introduction, the objectives are stated clearly: explore in-hospital  of asthma and COPD exacerbations in low-income nations.

Comments:

1) Title- while this study is from Jimma Ethiopia, the authors may want to re-consider their title and add Ethiopia after Jimma.

2) Sample size- the authors chose a 6 month study period with a relatively small 'N" of 130, did the authors consider extending the study period to 12 months to capture more patients.

3) Grammatical errors: There are numerous speling, syntax and grammatical errors throughout the manuscript which I have noted below.

3a) Abstract in methods: "Kaplan-Mayer  (Log-rank test) and Cox regression" The crrect spelling is Kaplan Meier.

3b) Abstract: "The total proportion of in-hospital mortality was 14(10.78%), making the incidence  rate of mortality, 2.56 per 1000 person-years; 95% CI [1.518-4.328]." The authros should re-word this sentence as it is unclear as written.

6. PLOS authors have the option to publish the peer review history of their article (what does this mean?). If published, this will include your full peer review and any attached files.

Reviewer #1: No

---

## [Author Response · Author response to Decision Letter 0]

15 Jun 2020

Dear editor, this is a rebuttal letter concerning the manuscript titled

“Mortality and its predictors among patients admitted with acute exacerbation of chronic obstructive respiratory diseases in Jimma Medical Centre: Prospective observational study” which you have requested for a response to the issues raised during the review process. Based on the comments given, all the authors have discussed on it and we have made a significant change to the manuscript, including the title (page 1). So, this is the highlight of what has been done. To make things easy, we kept all the concerns you raised unedited and provided our responses following each question. 

1. Question 1 is about the format,

Response: it was considered. (Page 1 to the final)

2. Thank you for including your ethics statement: The full protocol of this study was submitted for Institutional Review Board (IRB: IHRPGD/549/109) of JUMC, for ethical approval. After informing the overall concern of the study and confidentiality, written informed consent was taken written letter sheets; for those patients with problem of speaking, consent was requested from relatives. 

Response: This section is re-written and included under the section of “Study design, population and sample size determination” on page 5 of the manuscript.

3. Please correct your reference to "p=0.000" to "p<0.001" or as similarly appropriate, as p values cannot equal zero.

 Response: We have corrected it

4. Please include additional information regarding the survey or questionnaire used in the study and ensure that you have provided sufficient details that others could replicate the analyses. For instance, if you developed a questionnaire as part of this study and it is not under a copyright more restrictive than CC-BY, please include a copy, in both the original language and English, as Supporting Information. Moreover, please include more details on how the questionnaire was pre-tested, and whether it was validated.

 Response: The questionnaire was sent to you and the data collection tool validation was provided under the sub-heading of “Tool validation and data quality assurance” on the page 7 of the manuscript.

Response: Here I mean that the data will be sent to the journal if the manuscript is accepted for publication. So, you can directly access the data from the corresponding author if the manuscript is manuscript be accepted for publication.

6. Please amend either the abstract on the online submission form (via Edit Submission) or the abstract in the manuscript so that they are identical.

Response: Done

7. Your ethics statement must appear in the Methods section of your manuscript. If your ethics statement is written in any section besides the Methods, please move it to the Methods section and delete it from any other section. Please also ensure that your ethics statement is included in your manuscript, as the ethics section of your online submission will not be published alongside your manuscript.

Response: placed under method section on page 5 of the manuscript 

Reviewer's Responses to Questions

Dear reviewers, your comments are fully accepted and we have addressed as follow

Reviewer #1: The manuscript, "Mortality and its predictors among patients admitted with acute exacerbations of chronic obstructive respiratory diseases in Jimma Medical Center: Prospective cohort study", by T. Mega and colleagues reports the outcomes of a cohort of patients admitted with COPD exacerbations to a referral hospital in a low-resource country. The study is descriptive, but still adds to our understanding of the outcomes for COPD patients in different countries. However, the study has several important limitations, which are outlined below:

1. There are several grammatical and spelling errors throughout the manuscript, which makes the manuscript difficult to read. For example, the third sentence of the fourth paragraph of the Background reads, "An indoor (like smoking), outdoor air pollution and occupational asthma gens were risks quantified for asthma in GBD". This sentence appears to be missing a word or phrase. Additionally, "gens" is a mistake, and the authors' intent is not clear regarding what word is meant to be in this sentence. The manuscript should be carefully reviewed and these errors corrected.

Response: Thank you for your concern indeed. All your comments are constructive and we reviewed the manuscript for the English grammar, sentence structures and punctuation. We consulted English language and literature department of Jimma University and included all the corrections they forward. For the word “asthmagens- any substance that is causally-related to the development of asthma symptoms” is a single word which was written as two separate words, “asthma gens.” We have corrected it in the manuscript. 

2. The study population is very small given that the hospital serves a population of 15 million. The authors should provide reasons for this.

Response: Dear reviewer, we also acknowledge your concern regarding the sample size and that was why we have stated it as one of the major limitation. However, we would like to remind you that, this region had 4 additional referral hospitals and 20 general hospitals to which the cases can be admitted and managed. Jimma medical centre (a referral hospital with specialized care) is the highest level care in the region and it receives only patients referred from the aforementioned settings with the need of advanced medical care. So, it is likely that adequate sample size recruitment may need more than 6-months. Moreover, the authors didn’t get any funding support, one of hurdles for not extending the study period beyond six month. They covered all the data collection and investigational fees from their pocket. Because of this hardship, we were unable to prolong the study period. 

3. The authors should provide specific criteria as to how the underlying diagnosis of asthma or COPD was made in the cohort patients.

Response: The comments is highly appreciated and we included this under the sub-heading of “Diagnostic procedures for acute exacerbations of chronic respiratory diseases” on the page 8 of the manuscript.

4. The authors should provide some context for the demographic information of their cohort. Specifically, they should indicate whether there are specific differences between the patients and the rest of the communities served by Jimma Hospital, or if the patient cohort is generally representative of the surrounding areas.

Response: As mentioned above, Jimma medical centre delivers the most advanced level of care and it served as a referral centre for selected paints. It has better diagnostic facility, human capacity and medication services. This is the final level of the health care system (tertiary care hospital) in Ethiopia. Cases are being managed here those who can’t be managed at the previously mentioned hospitals or those needing further investigation and case management. 

5. There should be more details about the in-patient therapy received by the study cohort. Did any of the patients receive mechanical ventilation? What was the dose range and duration of steroid therapy used for treating the COPD/asthma exacerbations? This information will allow the interested reader to understand the differences in therapy received by the patients in the Jimma cohort and other COPD populations.

Response: Patients receiving mechanical ventilation were not included as they are not stable enough to provide lung function test. There were actually 3 patients receiving mechanical ventilation during this study and excluded. The details of the inpatient medication including corticosteroids for asthma and COPD were presented (Annex 1) .

6. The authors should summarize the post-hospital follow up provided to the patients. Was there a systematic post-discharge educational program given to the patients? Is there a community-based nursing follow up available?

Response: the highlight of the post-hospital care was provided under the sub-heading of “Description of Medications and investigational findings (page 14)” and there is no-community based nursing home. Hence, they will be linked to the OPD of JMC pulmonary clinic for regular follow-up.

7. The finding that patients with comorbid disease were less likely to die during follow up compared to patients without comorbidities is unexpected. The authors should enlarge their Discussion of this finding to include their assessment as to the reasons for this.

Response: we ask an apology for the fatal error we committed during the data analysis. As you saw this finding was contradictory not only with existing studies, but also with the descriptive section too. i.e 9 of the 14 deaths had comorbidity, and comorbidity become protective. With this controversial finding, thank you for your insight, we recalled the statistician to re-analyse the data and found that the variable “co-morbidity” should have been removed earlier as its binary cox regression p-value was 0.63. There are also other variables with similar fate. Thus, we revised this section and provided a new table with some modification. For the sake of convenience we have sent you the final model of the STATA output. 

Reviewer 2

This is an interesting study and the objective of the study are well stated. In the introduction, the objectives are stated clearly: explore in-hospital of asthma and COPD exacerbations in low-income nations.

1) Title- while this study is from Jimma Ethiopia, the authors may want to re-consider their title and add Ethiopia after Jimma.

Response: Done!

2) Sample size- the authors chose a 6 month study period with a relatively small 'N" of 130, did the authors consider extending the study period to 12 months to capture more patients.

• Response: Dear reviewer, we also acknowledge your concern regarding the sample size and that was why we have stated it as one of the major limitation. However, we would like to remind you that, this region had 4 additional referral hospitals and 20 general hospitals to which the cases can be admitted and managed. Jimma medical centre (a referral hospital with specialized care) is the highest level care in the region and it receives only patients referred from the aforementioned settings with the need of advanced medical care. So, it is likely that adequate sample size recruitment may need more than 6-months. Moreover, the authors didn’t get any funding support, one of hurdles for not extending the study period beyond six month. They covered all the data collection and investigational fees from their pocket. Because of this hardship, we were unable to prolong the study period.

3) Grammatical errors: There are numerous spelling, syntax and grammatical errors throughout the manuscript which I have noted below.

Response: Corrections made

3a) Abstract in methods: "Kaplan-Mayer (Log-rank test) and Cox regression" The crrect spelling is Kaplan Meier.

• Response: Corrected

3b) Abstract: "The total proportion of in-hospital mortality was 14(10.78%), making the incidence rate of mortality, 2.56 per 1000 person-years; 95% CI [1.518-4.328]." The authors should re-word this sentence as it is unclear as written.

Response: Re-written

---

## [Decision Letter · Decision Letter 1]

2 Jul 2020

PONE-D-20-02427R1

Mortality and its predictors among patients treated for acute exacerbations of chronic obstructive respiratory diseases in Jimma Medical Center; Jimma, Ethiopia: Prospective observational study

PLOS ONE

Dear Dr. Ayele,

Thank you for submitting your manuscript to PLOS ONE. After careful consideration, we feel that it has merit but does not fully meet PLOS ONE’s publication criteria as it currently stands. Therefore, we invite you to submit a revised version of the manuscript that addresses the points raised during the review process.

ACADEMIC EDITOR: I have received the comments of the reviewers on your manuscript. The specific comments of the reviewers are included below. Please provide point by point response in your revised manuscript.

We look forward to receiving your revised manuscript.

Kind regards,

Muhammad Adrish

Academic Editor

PLOS ONE

Reviewers' comments:

Reviewer's Responses to Questions

**Comments to the Author**

1. If the authors have adequately addressed your comments raised in a previous round of review and you feel that this manuscript is now acceptable for publication, you may indicate that here to bypass the “Comments to the Author” section, enter your conflict of interest statement in the “Confidential to Editor” section, and submit your "Accept" recommendation.

Reviewer #1: (No Response)

2. Is the manuscript technically sound, and do the data support the conclusions?

Reviewer #1: Partly

3. Has the statistical analysis been performed appropriately and rigorously? 

Reviewer #1: Yes

4. Have the authors made all data underlying the findings in their manuscript fully available?

Reviewer #1: No

5. Is the manuscript presented in an intelligible fashion and written in standard English?

Reviewer #1: No

6. Review Comments to the Author

Reviewer #1: The revised manuscript by T. Mega and colleagues has been much improved by the authors’ revisions. However, this study continues to have significant limitations, which are summarized below:

1. While improved, the writing continues to have many spelling, grammar, and word choice errors. Examples of these include:

a. Line 108 “cathment” instead of “catchment”

b. Line 120 “miss diagnosed” instead of “misdiagnosed”

c. Line 169 “throghly” instead of “thoroughly”

d. Line 169 “discudssed” instead of “discussed”

e. Line 234 “Moreover” is awkward.

f. Line 247 “peviuos” instead of “previous”

g. Line 258 “chronic comorbidities and it contributed” - the subject of “contributed” should be plural

h. Line 337 “predictor” instead of “predictors”

i. Line 393 “prevents” is not the correct word to describe an inverse association between comorbid disease and mortality.

The authors need to have the manuscript reviewed again in detail to correct these English errors

2. The description of the study population would be improved by providing additional context for the interested reader. For example, how does the high prevalence of smoking (21.5%) in the study population compare to the overall prevalence in Ethiopia? Approximately half of the subjects were from rural areas – does this reflect the overall population in the hospital catchment area? Additionally, since a substantial percentage (56%) of the study subjects did not have formal education, what percentage of the general population in the hospital catchment area and/or Ethiopia similarly does not have formal education?

3. The authors should provide additional information about the use of antibiotics for the patients, particularly since their use correlated with mortality. Since the subjects were all admitted with exacerbations of lung disease, the number of subjects who did not receive antibiotics is high. Are there hospital guidelines regarding the use of antibiotics? If not, is there a consensus among the practitioners in the hospital regarding the use of antibiotics for patients admitted with exacerbations of lung disease? Is the use of antibiotics likely to be a marker of disease severity? Why did the authors not include statistics regarding antibiotic use?

4. The Discussion still includes a statement (line 393) that the presence of comorbid disease was protective. However, the findings reported in the Results indicate that pulmonary hypertension, heart failure, and renal dysfunction were all associated with increased mortality. This discordance should be resolved.

5. The authors need to expand their discussion of the study limitations. The small sample size resulted in the study lacking statistical power. The study therefore likely missed predictors of mortality. An example of this would be the lack of association between prior hospitalization and mortality. The authors need to more fully outline this limitation of their study. Additionally, the exclusion of patients requiring mechanical ventilation needs to be described as a limitation. Although the authors provide the rationale for this exclusion, patients who require mechanical ventilation for an exacerbation of lung disease have a high in-hospital mortality. Thus, excluding this population limited the study’s ability to detect risk factors for in-hospital mortality.

6. Table 5 has an asterisk after the number of night attacks. It isn't clear if this refers to something that should have additional clarification in the Table legend or if this is a typographical error. The authors should either clarify the use of the asterisk or remove it.

7. PLOS authors have the option to publish the peer review history of their article (what does this mean?). If published, this will include your full peer review and any attached files.

Reviewer #1: No

---

## [Author Response · Author response to Decision Letter 1]

15 Aug 2020

Response to Reviewers'

Dear reviewers, the authors will fully acknowledge the importance of your feedbacks to produce a scientifically sound manuscript. Therefore, we incorporated them to the manuscript given they are directly related to the objectives defined by the investigators. Other points intended for explanation were described on the rebuttal letter only. The comments were addressed in the following manner; keeping all statements of the reviewers as they were. 

Reviewer #1: 

The revised manuscript by T. Mega and colleagues has been much improved by the authors’ revisions. However, this study continues to have significant limitations, which are summarized below:

1. While improved, the writing continues to have many spelling, grammar, and word choice errors. Examples of these include:

a. Line 108 “cathment” instead of “catchment”

b. Line 120 “miss diagnosed” instead of “misdiagnosed”

c. Line 169 “throghly” instead of “thoroughly”

d. Line 169 “discudssed” instead of “discussed”

e. Line 234 “Moreover” is awkward.

f. Line 247 “peviuos” instead of “previous”

g. Line 258 “chronic comorbidities and it contributed” - the subject of “contributed” should be plural

h. Line 337 “predictor” instead of “predictors”

i. Line 393 “prevents” is not the correct word to describe an inverse association between comorbid disease and mortality.

The authors need to have the manuscript reviewed again in detail to correct these English errors

Response: The whole document was fully revised for these and other errors.

2. The description of the study population would be improved by providing additional context for the interested reader. For example, how does the high prevalence of smoking (21.5%) in the study population compare to the overall prevalence in Ethiopia? 

Response: Though no recent data regarding the national prevalence of smoking, the 2015 published study by WHO indicated 8.9% and 0.5% of Ethiopian men and women, respectively, were smokers, describing it as an ongoing and direct public health threat. (https://tobaccoatlas.org/wp-content/uploads/pdf/ethiopia-country-facts.pdf). In the context of this manuscript, such high prevalence of smoking may not be surprising as majority of the study participants were COPD (71/130) patients and smoking is the well-known risk factor. 

Approximately half of the subjects were from rural areas – does this reflect the overall population in the hospital catchment area? 

Response: Yes! The rural population in Ethiopia was 78.78 % in 2019 (World Bank). So, this catchment area also mirrors similar trend. 

Additionally, since a substantial percentage (56%) of the study subjects did not have formal education, what percentage of the general population in the hospital catchment area and/or Ethiopia similarly does not have formal education?

Response: The national literacy rate as reported by World Bank in 2017 was 52% (https://data.worldbank.org/indicator/SE.ADT.LITR.ZS). So, the data reflects similar fact. 

3. The authors should provide additional information about the use of antibiotics for the patients, particularly since their use correlated with mortality. Since the subjects were all admitted with exacerbations of lung disease, the number of subjects who did not receive antibiotics is high. Are there hospital guidelines regarding the use of antibiotics? If not, is there a consensus among the practitioners in the hospital regarding the use of antibiotics for patients admitted with exacerbations of lung disease? Is the use of antibiotics likely to be a marker of disease severity? Why did the authors not include statistics regarding antibiotic use?

Response: Dear reviewer, it is a global fact that all patients admitted with acute exacerbations of asthma and COPD must receive a course of antibiotics. This data was annexed as inpatient medication in the table named “annex A” in our manuscript after table of contents. If you see the data regarding inpatient medications in this table, the number of patients exposed to antibiotics was far beyond the study population, as a patient might receive one or more antibiotics. 

Regarding exclusion of antibiotics data from statistics section: the reviewers should note that not only antibiotics but also other several variables were removed because of the fact that either the variables were failed on cell adequacy test or they were removed from the model due to their p-value >0.25 on binary logistic regression analysis.

4. The Discussion still includes a statement (line 393) that the presence of comorbid disease was protective. However, the findings reported in the Results indicate that pulmonary hypertension, heart failure, and renal dysfunction were all associated with increased mortality. This discordance should be resolved.

Response: This expression was erroneous and corrected according to the finally corrected data. You can also look at the STATA output (the final model) attached as supplementary data in our previous revision sent to PLOS ONe.

5. The authors need to expand their discussion of the study limitations. The small sample size resulted in the study lacking statistical power. The study therefore likely missed predictors of mortality. An example of this would be the lack of association between prior hospitalization and mortality. The authors need to more fully outline this limitation of their study. Additionally, the exclusion of patients requiring mechanical ventilation needs to be described as a limitation. Although the authors provide the rationale for this exclusion, patients who require mechanical ventilation for an exacerbation of lung disease have a high in-hospital mortality. Thus, excluding this population limited the study’s ability to detect risk factors for in-hospital mortality.

Response: We included these precious statements in our limitation section. Thank you indeed!

6. Table 5 has an asterisk after the number of night attacks. It isn't clear if this refers to something that should have additional clarification in the Table legend or if this is a typographical error. The authors should either clarify the use of the asterisk or remove it.

Response: Dear reviewer, these are not asterisk, by to say “times” (sign of multiplication). Anyhow removing the asterisk will not affect the content of the information and so, we have removed them.

---

## [Decision Letter · Decision Letter 2]

31 Aug 2020

Mortality and its predictors among patients treated for acute exacerbations of chronic obstructive respiratory diseases in Jimma Medical Center; Jimma, Ethiopia: Prospective observational study

PONE-D-20-02427R2

Dear Dr. Ayele,

We’re pleased to inform you that your manuscript has been judged scientifically suitable for publication and will be formally accepted for publication once it meets all outstanding technical requirements.

Kind regards,

Muhammad Adrish

Academic Editor

PLOS ONE

Additional Editor Comments (optional):

Reviewers' comments:

Reviewer's Responses to Questions

**Comments to the Author**

1. If the authors have adequately addressed your comments raised in a previous round of review and you feel that this manuscript is now acceptable for publication, you may indicate that here to bypass the “Comments to the Author” section, enter your conflict of interest statement in the “Confidential to Editor” section, and submit your "Accept" recommendation.

Reviewer #1: All comments have been addressed

2. Is the manuscript technically sound, and do the data support the conclusions?

Reviewer #1: (No Response)

3. Has the statistical analysis been performed appropriately and rigorously? 

Reviewer #1: (No Response)

4. Have the authors made all data underlying the findings in their manuscript fully available?

Reviewer #1: (No Response)

5. Is the manuscript presented in an intelligible fashion and written in standard English?

Reviewer #1: (No Response)

6. Review Comments to the Author

Reviewer #1: (No Response)

7. PLOS authors have the option to publish the peer review history of their article (what does this mean?). If published, this will include your full peer review and any attached files.

Reviewer #1: No

---

## [Editor Report · Acceptance letter]

9 Sep 2020

PONE-D-20-02427R2 

Mortality and its predictors among patients treated for acute exacerbations of chronic obstructive respiratory diseases in Jimma Medical Center; Jimma, Ethiopia: Prospective observational study 

Dear Dr. Mega:

I'm pleased to inform you that your manuscript has been deemed suitable for publication in PLOS ONE. Congratulations! Your manuscript is now with our production department. 

Kind regards, 

on behalf of

Dr. Muhammad Adrish 

Academic Editor

PLOS ONE